# Analysis of PD-L1 and CD3 Expression in Glioblastoma Patients and Correlation with Outcome: A Single Center Report

**DOI:** 10.3390/biomedicines11020311

**Published:** 2023-01-22

**Authors:** Navid Sobhani, Victoria Bouchè, Giovanni Aldegheri, Andrea Rocca, Alberto D’Angelo, Fabiola Giudici, Cristina Bottin, Carmine Antonio Donofrio, Maurizio Pinamonti, Benvenuto Ferrari, Stefano Panni, Marika Cominetti, Jahard Aliaga, Marco Ungari, Antonio Fioravanti, Fabrizio Zanconati, Daniele Generali

**Affiliations:** 1Department of Medicine, Section of Epidemiology and Population Sciences, Baylor College of Medicine, Houston, TX 77030, USA; 2Department of Medical, Surgery and Health Sciences, University of Trieste, 34147 Trieste, Italy; 3Department of Biology & Biochemistry, University of Bath, Bath BA27AY, UK; 4Neurosurgery, ASST Cremona, Viale Concordia 1, 26100 Cremona, Italy; 5Division of Biology and Genetics, Department of Molecular and Translational Medicine, University of Brescia, Viale Europa 11, 25123 Brescia, Italy; 6Breast and Brain Unit, ASST Cremona, Viale Concordia 1, 26100 Cremona, Italy; 7Pathology Unit, ASST Cremona, Viale Concordia 1, 26100 Cremona, Italy

**Keywords:** PD-L1, CD3, biomarkers, immune checkpoint inhibitors, glioblastoma

## Abstract

With the advent of immunotherapies, the field of cancer therapy has been revived with new hope, especially for cancers with dismal prognoses, such as the glioblastoma multiforme (GBM). Currently, immunotherapies should potentiate the host’s own antitumor immune response against cancer cells, but it has been documented that they are effective only in small subsets of patients. Therefore, accurate predictors of response are urgently needed to identify who will benefit from immune-modulatory therapies. Brain tumors are challenging in terms of treatments. The immune response in the brain is highly regulated, and the immune microenvironment in brain metastases is active with a high density of tumor-infiltrating lymphocytes (TILs, CD3+ T cells) in certain patients and, therefore, may serve as a potential treatment target. In our study, we performed immunohistochemistry for CD3 and PD-L1 along the routine assessment of the O6-methylguanine-methyltransferase (MGMT) promoter methylation status and the IDH1 and 2 status in a single center cohort of 69 patients with GBM (58 primary tumors and 11 recurrences) who underwent standard multimodal therapies (surgery/radiotherapy/adjuvant temozolamide). We analyzed the association of PD-L1 tumor expression and TILs with overall survival (OS). The PD-L1 expression was observed in 25 of 58 (43%) newly diagnosed primary glioblastoma specimens. The sparse-to-moderate density of TILs, identified with CD3+ expression, was found in 48 of 58 (83%) specimens. Neither PD-L1 expression nor TILs were associated with overall survival. In conclusion, TILs and/or PD-L1 expression are detectable in the majority of glioblastoma samples, and even if they slightly relate to the outcome, they do not show a statistically significant correlation.

## 1. Introduction

Glioblastoma multiforme (GBM) is the most frequent brain tumor in adult individuals and is correlated to the dismal median overall survival rate of approximately 15 months, mainly due to its rapid and extremely invasive growth rate, with a tendency to infiltrate and destroy brain tissue [1].

Maximal safe resection followed by Temozolomide and radiation therapy is currently the first choice for patients diagnosed with GBM [2,3]. Despite the initial response to therapy, most patients experience disease progression or early tumor recurrence, which is rapidly lethal [4]. Several experimental treatments, including inhibitors of specific cell-cycle pathways (e.g., everolimus, gefitinib, erlotinib, imatinib), and antiangiogenic agents (e.g., bevacizumab) have so far resulted in limited survival benefits for those diagnosed with GBM, and a new treatment approach is needed [5].

GBM is largely considered an immunosuppressive tumor due to the activation of different immune escape strategies such as the upregulation of programmed death ligand 1 (PD-L1), indolamine 2,3 dioxygenase (IDO) and transforming growth factor-β (TGF-β) [6]. However, the central nervous system (CNS) is no longer considered an immunologically privileged site. Resident macrophages, lymphatic vessels, and dural sinuses have been discovered. Additionally, it has been shown that there are active interactions between CNS and peripheral immune cells [7]. Cumulating evidence indicates that tumor-infiltrating lymphocytes (TILs) can penetrate GBM [8].

The more recent discovery of immune checkpoint inhibitors (ICIs), a novel group of molecules with antitumor activity, has been reported as a significant turning point in immuno-oncology with durable tumor remission and significant response rates [9]. These molecules—generally in the form of monoclonal antibodies—act by blocking immunosuppressive T cell-surface receptors (e.g., PD-1, CTLA-4) that switch off effector T cells, thus resulting in the antitumor immune response. Several ICIs have been tested against PD-1 (Pembrolizumab, Nivolumab, Cemiplimab), its ligand PD-L1 (Atezolizumab, Avelumab, Durvalumab) and CTLA-4 (Ipilimumab) [10,11]. Clinical trials testing these drugs have shown promising therapeutic responses in some specific cancers, such as melanoma, clear cell renal carcinoma, Non-Small Cell Lung Cancer (NSCLC), triple-negative breast cancer, head and neck squamous cell carcinoma, Hodgkin’s lymphoma, bladder carcinoma, hepatocarcinoma, and Merkel cell carcinoma and favorable toxicity profile across tumor types [12,13]. Many of these drugs have been approved by the Food and Drugs Administration in patients with advanced or unresectable tumors or as second-line therapy [12,14].

In more detail, PD-1 is a cell surface co-inhibitory receptor expressed on CD8+ T cells which block the immune response when bound to PD-L1—expressed on cancer cells—thus leading to T cell “exhaustion” [15]. High PD-L1 expression levels on tumor cells seem to correlate to a therapeutic benefit from using ICIs targeting the PD-1/PD-L1 axis, as suggested by new evidence from preclinical studies and experimental models [16,17,18]. Ultimately, while the absolute number of TILs appears to be a less strong predictor in patients with GBM, emerging evidence is observing a significant activation of PD-L1 in GBM that might emerge as a clinically relevant biomarker [8,19]. Although with controversial results, several studies have investigated the correlation between prognosis and PD-L1 and CD3 biomarkers expression. Of note, Wang et al. performed a meta-analysis of 15 studies, including a total of 108 patients. The outcome reported that increased PD-L1 expression can predict unfavorable OS in GBM, therefore corroborating the idea of PD-L1 as a prognostic biomarker for GBM [20].

Similarly, El Samman et al. performed immunohistochemical analysis on 30 GBM cancer samples reporting that high PD-L1 expression was correlated with worse PFS and OS where the correlation between CD3 and prognosis was statistically insignificant [21]. The systematic literature review and meta-analysis by Hao and colleagues corroborated the concept that PD-L1 is correlated with poor survival, thus exerting the role of a prognostic marker [20]. Indeed, the analysis involved 9 studies for a total of 806 GBM patients. The pooled results reported a significant correlation between poor survival and PD-L1 expression.

In contrast, only a few investigations reported no significant correlation between prognosis and expression of PD-L1 and CD3 markers. Berghoff et al. assessed CD3 and PD-L1 expression in 135 GBM samples using an immunohistochemistry assay. Although PD-L1 and CD3 markers were significantly detectable in most GBM, these were not correlated with prognosis [22]. Ultimately, we reported the study by Dede and colleagues where 90 GBM cancer samples were immunohistochemically investigated. A slightly positive correlation was found between CD3 expression and better prognosis, whereas no significant correlation was found between PD-L1 expression and worse prognosis [23].

Here, we present an observational, non-interventional retrospective clinical investigation analyzing the presence of CD3 and PD-L1 (pan-TILs index) in a group of 69 patients with GBM. Neither PD-L1 expression nor tumor lymphocyte infiltrate was associated with overall survival in this cohort of patients.

## 2. Patients and Materials

### 2.1. Cremona Retrospective Cohort

We retrospectively identified formalin-fixed paraffin-embedded (FFPE) tumor tissues specimens of 69 adult patients who underwent neurosurgical resection of newly diagnosed GBM at the Department of Neurosurgery of the ASST Cremona, between 2018 and 2020 from the Neuro-Biobank archive. From this cohort, only 58 patients who originated as primary tumors were included in the analyses.

The histological diagnosis of GBM was conducted according to the current WHO classification [24], thus excluding cases with isocitrate dehydrogenase (IDH) mutation. The clinical and survival data were retrieved from the chart review.

The ethics committee at the ASST of Cremona approved the study (protocol number 32219, dated 2 October 2019). The characteristics of patients, including clinical baseline parameters, MGMT promoter methylation status results, MIB1, P53, PD-L1, and CD3 levels, are listed in Table 1.

### 2.2. Immunohistochemical Staining

In order to perform immunohistochemical staining, FFPE blocks were cut into serial 3 μm thick slices with a microtome. Immunohistochemistry for CD3 and PD-L1 has been performed with a Ventana Benchmark Ultra immunostaining automated system. Immunostaining was performed on adjacent sections to facilitate a comparison of regional distributions of PD-L1 and CD3 expression and TIL presence. Antibodies and immunostaining protocols are listed in Appendix A. FFPE tissue blocks of human non-neoplastic lymph nodes (TILs) and human placenta (PD-L1) were used as positive controls.

### 2.3. Evaluation of Immunohistochemistry

#### 2.3.1. PD-L1 Expression

The PD-L1 expression was recorded in accordance with cellular and topographical distribution, localization, and intensity of the immunohistochemical signal. A semiquantitative assessment for the extent of diffuse/fibrillary PD-L1 was conducted. The presence of epithelioid tumor cells comprising distinct membranous anti-PD-L1 labeling was registered only if they accounted for >5% of all tumor cells through the semiquantitative assessment to comply with the cutoff of previous publications [22]. The tumor proportion score (TPS) was assessed, estimating the number of tumor cells with PD-L1 cytoplasmic expression of any intensity over the total number of tumor cells expressed in percentage. Cut-offs of 1% and 50% were used to define cases with minimal/absent (staining in <1% of tumor cells), intermediate (staining in ≥1% and <50% of tumor cells), and high (≥50%) PD-L1 expression. Cases with minimal/absent expression were considered PD-L1-negative, while those with intermediate and high expression were considered PD-L1-positive.

#### 2.3.2. Tumor-Infiltrating Lymphocyte Density

The density of CD3+ TIL subset was evaluated by overall impression at low microscopic magnification (10×; Leica DM 2500 LED) to indicate, by visual estimation, whether CD3-positive lymphocytic infiltrate was present and its intensity (“absent”, “mild”, “moderate/high”). Furthermore, the densities were separately scored at higher magnification (40×) by counting positive cells in three separate representative microscopic fields and dividing the total by three. Previously published semiquantitative evaluation criteria were used to describe TIL density [25], which was judged as lower and higher than the median. The TIL distribution and PD-L1-positive tumor areas were compared on adjacent tissue sections. Necrotic and perivascular areas were excluded from the evaluation.

### 2.4. MGMT and MIB-1

The immunohistochemical methodology used in Cremona for routine markers is thoroughly described elsewhere [26]. Briefly, an antigen retrieval step was performed by heating a tissue section in a citrate buffer.

The following cut-offs were considered for studying the methylation status of MGMT: 0–10%: unmethylated; 10–29%: methylated; >30%: hypermethylated. E3 ubiquitin-protein ligase (MIB-1) is a marker of proliferation in various cancers. We explored this marker to predict survival in our 69 GBM cohort. The analysis was conducted both considering MIB-1 as a continuous variable and as a binary variable with a cut-off of 35%, corresponding to its median value.

### 2.5. p53

Pyrosequencing analysis was used to detect p53 mutations. The Therascreen *MGMT* Pyro Kit and the PyroMark Q24 system (both from Qiagen, Germany) were used to assess the methylation status of the *MGMT* gene promoter. In brief, bisulfite-converted genomic DNA was amplified by PCR, the amplicons were immobilized on streptavidin beads, and single-stranded DNA was prepared, sequenced, and finally analyzed on the PyroMark Q24 system. p53 mutations were further validated by immunohistochemical procedures [26]. The primary antibodies were applied for p53 (mouse monoclonal D07, Leica Biosystems) at a dilution of 1:100, 1 h incubation at room temperature.

### 2.6. Statistical Analysis

Because of the low number of observations in some subgroups and the non-normality of the distribution of variables (assessed with the Shapiro-Wilk test), comparisons among groups were performed with the Fisher exact test for proportions and Wilcoxon rank sum test with continuity correction for numeric variables. Spearman’s rank correlation was used to analyze the correlations between numerical variables. The Kaplan-Meier product limit method was used to estimate the overall survival of patients with glioblastoma from the date of surgery to death or last follow-up. Group differences were assessed using the log-rank test. The association of patient and tumor variables with survival was tested with Cox regression models. The proportionality of hazards was assessed with Schoenfeld residuals, while Martingale and deviance residuals were used to assess nonlinearity and influential observations, respectively. Some variables with strong evidence of non-proportionality of hazards were modeled with time-dependent coefficients (tt function of r survival package). Variables with Wald p-values <0.05 at univariate Cox models (adding time-dependent coefficients when needed) were included in a multivariate analysis. With a backward process, variables were removed when this led to a reduced Akaike information criterion (AIC). Analyses were conducted with R 4.2.1.

## 3. Results

### 3.1. Cremona Retrospective Cohort

#### 3.1.1. Patients

Figure 1 shows the characteristic morphology of glioblastoma with palisade necrosis (Figure 1a) and microvascular proliferation (Figure 1b).

Patient and tumor characteristics are reported in Table 1. Of the 69 evaluable cases, 58 are from primary tumor samples operated on at the clinical onset of disease, while 11 specimens are from surgical resections of tumor recurrences. There are no matched cases of primary tumor and recurrence. The median age of the entire case series is 64 years (range 41–81), with a peak incidence between the ages of 70 and 75 years (Appendix A). The brain-site distribution of glioblastomas for the entire cohort of patients is illustrated in Appendix A. Among primary tumors, 26 (45%) involved the frontal lobe, 10 (17%) the temporal lobe, 4 (7%) the parietal lobe, and 2 (3%) the occipital lobe, while the remaining cases were at the border between different lobes. Fifty-five percent of the primary tumors involved the right hemisphere and 45% involved the left one. Thirty percent of the patients with a primary tumor underwent gross total resection (GTR), 27% a subtotal resection, and 43% a partial resection. About 80% of the primary tumors were unifocal and 20% multifocal.

Among 65 samples evaluable for MGMT promoter methylation in the entire cohort, 37 were unmethylated (56.9%), 10 methylated (15.4%), and 18 hypermethylated (27.7%) (Appendix A). The MGMT promoter was methylated (≥10% of tumor cells) in 38% of primary tumors and 78% of recurrences, and this difference is statistically significant (Fisher exact test *p*-value = 0.03). Median Mib1 does not differ significantly between primary tumors and recurrences. The distribution of p53 status showed a non-significant difference between primary tumors and recurrences (Fisher’s exact test *p*-value = 0.059 when comparing the three categories of positive, intermediate, and negative p53; *p*-value = 0.08 when considering the intermediate category as wild type and the negative plus positive cases as mutated).

#### 3.1.2. PD-L1 Expression

Different expression patterns of PD-L1 were observed in the glioblastoma tissues *(*Figure 2). In the whole case series, 7 (10%) glioblastomas showed “high” (staining in ≥50% of tumor cells) PD-L1 expression, and 21 (31%) showed intermediate (staining in 1% to 50% of tumor cells) expression, for a total rate of PD-L1-positive cases of 41% (Figure 3a). When considering only the 58 primary tumor cases, 7 (12%) had high expression, and 18 (31%) had an intermediate expression of PD-L1, for a total rate of “positive” cases of 43%; 33 (57%) showed only focal/minimal expression of the marker (<1%), or did not express it at all, and were considered PD-L1-“negative”. Out of 11 recurrent cases, 8 (73%) were PD-L1-negative and 3 (27%) were PD-L1-positive. The difference in the rate of PD-L1 positivity between primary tumors and recurrences was not statistically significant, perhaps also due to the small number of recurrences (Fisher’s exact test *p* = 0.5).

#### 3.1.3. Tumor-Infiltrating Lymphocytes

TILs were evaluated with an antibody against CD3, directed to T-lymphocytes (Figure 4). Among the whole case series, in 44 patients (64%) the lymphocytic infiltrate was mild, in 13 patients (19%) it was moderate/high; and in the remaining 12 patients (17%) CD3 was not significantly expressed (Figure 3b and Figure 4b). Among the 57 patients expressing CD3 (both mild and moderate expression), 30 (53%) were negative for PD-L1, while 27 (47%) were positive. In the group of patients expressing PD-L1 (intermediate or high positivity), all cases had a mild to moderate CD3+ lymphocytic infiltrate. There was a significant positive correlation between CD3-positive TILs count and the percentage of PD-L1-positive tumor cells (Spearman’s rank correlation coefficient: 0.32; *p*-value: 0.0076; Figure 4c), and a significant positive association between CD3-positivity and PD-L1-positivity in tumor samples (Fisher’s exact test *p*-value = 0.02, Figure 3c).

When considering only primary tumors, CD3 expression was absent in 10 (17%), mild in 37 (64%), and moderate/high in 11 (19%) cases. Among the 48 CD3-positive tumors (CD3 mild or moderate/high), 24 (50%) were PD-L1-negative and 24 (50%) PD-L1-positive, while among the 10 CD3-negative cases, only one (10%) was PD-L1-positive and 9 (90%) PD-L1 negative. The difference in the rate of PD-L1-positivity between CD3-positive and CD3-negative cases is statistically significant (Fisher’s exact test *p* = 0.033), and there is a positive correlation between CD3-positive TILs count and the percentage of PD-L1-positive tumor cells within primary tumors (Spearman’s rank correlation rho 0.365, *p* = 0.005).

The CD3-positive TILs count did not differ significantly between primary tumors and recurrences, perhaps at least in part due to the small number of recurrent samples. Similarly, there was no correlation between CD3-positive TILs count and the percentage of PD-L1-positive tumor cells within the recurrent cases (Spearman’s rank correlation rho 0.07, *p* = 0.84).

### 3.2. Survival Analyses

#### 3.2.1. Association of Patient and Tumor Features with Survival

The median overall survival (OS) of the entire cohort of patients was 9 months (Appendix A), that for patients operated for primary tumors was 9.1 months (95% C.I. 6.9–11.9 months), and that for patients operated for tumor recurrence was 6.1 months (95% C.I. 3.06–not assessable) (Appendix A). The difference in OS between the two groups was not statistically significant, likely due to the small number of recurrent cases.

The results of univariate survival analysis in patients operated for the primary tumor are shown in Table 2. Due to non-proportional hazards, some variables have been modeled with time-dependent coefficients.

There was a statistically significant decrease in patients’ survival rates with growing age at diagnosis, although the impact of age diminished over time after surgery. Splitting the study population into two age-based groups, the median survival was 13.8 months in patients younger than 65 years (50%) and 7.4 months in patients aged 65 years or older (50%) (Figure 5a; log-rank test *p*-value = 0.0009).

The extent of tumor resection did also significantly influence survival. In the subgroup of patients who underwent partial resection, the average survival was 7.4 months. In patients with subtotal resection, it was 13.6 months, and in those with GTR 11.9 months (log-rank test *p*-value = 0.02).

The adverse impact of multifocality, absent immediately after surgery gradually becomes more pronounced with increasing time from surgery. Despite this, the median survival was comparable in unifocal and multifocal tumors (9.3 vs. 9.07 months) (Figure 5c).

A protective impact of MGMT promoter methylation, absent in the initial period, gradually emerged with increasing time elapsed since surgery (Figure 5d).

Sex, the involved hemisphere side, Mib1, and p53 status did not significantly impact OS.

#### 3.2.2. Survival Analyses in Relation to PD-L1 or CD3

In the PD-L1-negative group (33 patients), the median OS was 8.6 months. In the positive one (25 patients), it was 11.6 months.

Based on these analyses, PD-L1 does not seem to correlate with the overall survival rate of patients with GBM (*p*-value = 0.47; Figure 6a).

Similarly, the median OS was 8 months in patients whose tumors showed low (inferior to the median) CD3 expression versus 9.5 months in those with high expression (greater than or equal to the median) (log-rank *p*-value = 0.4, Figure 6b). When considering three levels of CD3 expression, namely absent, mild, and moderate/high, the median OS was 8.1 months in the CD3-absent group, 8.94 months in the mild CD3 group, and 12.0 months in the moderately/high CD3 group. However, this trend toward increased survival was not strong enough to produce a statistically significant result (*p*-value of 0.9).

A combination of the two variables PD-L1 and CD3, either their concomitant presence or the presence of at least one of them, was also not found to be associated with OS (data not shown).

We conducted a multivariate analysis by fitting into a Cox model all variables found to be significant at 0.05 in univariate analysis (adding time-dependent coefficients when indicated) and the variables relevant to the present study, namely PD-L1 (positive in ≥1% of tumor cells versus negative) and CD3 (expression above or equal to the median versus below the median). PD-L1 and CD3 are confirmed to be not associated with OS. At the same time, the other variables, that is, age, type of surgery, multifocality, and MGMT promoter methylation, are retained in the final model (Table 3).

## 4. Discussion

PD-1/PD-L1 interactions are now considered central immunological checkpoints of cancer. Here we detected clear membranous PD-L1 expression with different highlights of tumor cell surfaces, as seen in epithelial cancers and melanoma [27].

The rate of PD-L1-positive cases in glioblastoma in this study was comparable with other solid tumor types. It was expressed in about 30% of melanoma patients and 25–36% of non-small lung cancers, and 88% of glioblastomas by immunohistochemistry [28,29]. Our study corroborates previous publications indicating that the PD-1/PD-L1 axis plays an important role in creating an immunosuppressive microenvironment in glioblastoma. Overall, our data and the current experimental evidence provide a strong rationale for clinical trials to investigate the efficacy of checkpoint inhibitors for glioblastomas. In line with this, the latest reports indicate the durable antitumor efficacy of immune checkpoint inhibitors in animal glial brain tumor models [30,31]. Furthermore, the therapeutic efficacy of CTLA-4 antibody ipilimumab clinically evinced the feasibility of an antibody-mediated immune checkpoint inhibition in intraparenchymal CNS lesions [32,33]. On the other hand, there is a need for more studies to help clarify the capacity of these drugs to penetrate the blood–brain and blood–tumor barriers, essential for intracerebral delivery and efficacy of the immune checkpoint inhibitors for the CNS site. PD-1/PD-L1 should also be investigated in clinical trials of patients with gliomas.

CD3 defines the T-cell lineage. Here we confirm the presence of T-lymphocytic infiltrate in most glioblastoma patients (about 83%), which supports the hypothesis that GBM is not a cold tumor and suggests the presence of specific immunogenicity in cancer, as also highlighted by previous research publications [34,35,36,37,38]. This is an essential criterion for possible immunological treatment. The study by Kmiecik et al. demonstrated a correlation between the presence of CD3 and better survival if local lymphocytes could effectively stem the tumor or at least slow its growth [34]. Unfortunately, however, not much other evidence correlates CD3 with a more advantageous survival rate; indeed, its presence seems controversial in other studies. Song et al. [37] and Orrego et al. [38] observed that CD3 was a negative prognostic marker. Their reported patients had a shorter OS. In our study, however, the median OS showed a slight increase for CD3-positive cases.

In the future, a more specific analysis to assess the individual lymphocyte components could be beneficial to know if the T-cell population to the T-regulatory population ratio is altered in correlation with the survival of the patients. Sayour et al. conducted a survival analysis of 39 patients with primary and relapsing GBM. The percentages of CD3 were comparable to our study [39]. Additionally, they found that therapy-resistant relapsing patients had a higher FoxP3/CD3 ratio and a reduced CD4+/CD3+ and CD8+/CD3+. While a reduction in CD8+/CD3+ seems to have led to shorter survival, a high CD8+/CD3+ ratio was associated with better survival [39,40,41,42]. An interesting aspect in patients with moderate CD3 levels is that almost all are unmethylated MGMT. Regarding this, the distribution of MGMT in our population of patients is in accordance with other studies (Appendix A) [38,43,44].

Moreover, PD-L1 was present in about 40% of the patients recruited, and the data obtained from this study agree with those of other similar studies [8,43,44,45,46]. Its presence can mean either that the tumor is exploiting immune checkpoints to evade tumor surveillance or it can be an index of a high rate of lymphocyte depletion. Although, for the time being, ICIs therapy has not yielded the expected results in patients with GBM, deviating significantly from the benefit this therapy has brought in other cancers, PD-L1 expression is consistent in GBM. In Table 4, we have summarized ongoing phase II/III clinical trials investigating ICIs in glioblastomas. Interestingly, PD-L1 may be a valuable marker for selecting patients most likely to benefit from ICIs. The literature shows that high levels of PD-L1 in GBM are associated with a worse prognosis [46]. This is consistent with the downward-sloping attitude of the survival curve in this study, which showed that PD-L1 alone did not provide a statistically reliable result in our 58 primary tumor–patient cases (*p*-value of 0.47) [8,43,44,45].

In conclusion, from our study, PD-L1 alone or CD3 alone were not predictive of survival. PD-L1 is not alone as a predictor of survival but is considered a predictor of tumor evasion. Therefore, PD-L1 usefulness as a marker has been only that of predicting those patients who would best respond to anti-PD-L1 or anti-PD1 therapies.

Additionally, isocitrate dehydrogenase (IDH) is an enzyme catalyzing the oxidative decarboxylation of isocitrate to produce alpha-ketoglutarate and CO_2_. In brain tumors such as GBM, mutations of IDH1 are commonly correlated with more prolonged survival [47,48,49]. Regarding our IDH analysis, we observed a slight, non-significant, 3-month advantage in patients with an IDH-mutated variant, consistent with data from the scientific literature in which the IDH mutation has repeatedly been associated with more prolonged survival [50]. Nevertheless, the survival advantage of IDH mutant in GBM appears to be more significant (9.9 months vs. 24 months) in other related studies [51,52]. This could be due to the low representativeness of the IDH mutant population in this analysis, which was disproportionately low compared to its wild-type counterpart.

Another independent biomarker predictive of improved survival in GBM undergoing chemotherapy is the methylation status of the O6-methylglutanine-DNA methyl-transferase (MGMT) gene promoter [53]. As per our MGMT methylation analysis, the trend was in line with the literature showing an improvement of GBM with the methylation, although there are challenges in the literature regarding a consensus on the method of detection of this biomarker [53].

We recognize that our study’s sample size and retrospective nature limit the power of survival analyses in our study. The results should be confirmed in a larger and prospectively collected cohort. Additionally, the absence of information on PD-L1 and CD3 levels in response to treatment is another limitation to the study, which should be addressed in the future, especially concerning potential immunotherapies that other clinical trials could be investigating.

Finally, we analyzed the mutation of PD-L1 in TCGA samples from cBioportal. The analysis in 142 patients showed no correlation between PD-L1 and the survival of GBM patients. This agrees with our cohort of patients showing no correlation between the expression of PD-L1 and survival of GBM. Precisely, we obtained survival and PD-L1 gene (CD274) expression data (mRNA Expression, RSEM Batch normalized from Illumina HiSeq_RNASeqV2) for a cohort of 142 patients with IDH-wild type GBM from The Cancer Genome Atlas (TCGA) PanCancer Atlas [54] available at cBioPortal for Cancer Genomics [55]. In order to see if there was an association between PD-L1 expression levels and survival, we classified samples into two groups, with PD-L1 expression below or above the median. None of the patients for whom all these data are available had received therapies with immune checkpoint inhibitors. The results in Figure 7a show no statistically significant difference in OS between patients with PD-L1 expression below or above the median (log-rank test *p*-value = 0.736). Additionally, progression-free survival (PFS) did not differ significantly in the two groups, although there is a trend toward better PFS in patients whose tumors have PD-L1 expression below the median (log-rank test *p*-value = 0.0671, Figure 7b). When considering four quartiles of PD-L1 expression and comparing quartiles with each other or the first with the fourth quartile, results did not change.

## 5. Conclusions

In conclusion, our analysis has shown that neither CD3 nor PD-L1 expression correlate with improved glioblastoma survival. Additionally, the PD-L1 expression levels in 43% of primary glioblastoma patients suggest an immunosuppressive PD-1/PD-L1 active axis in a proportion of glioblastoma patients. Moreover, since PD-L1 has been used in solid tumors as a valid biomarker for a potential response to an immune checkpoint blockade against the PD-1/PD-L1 axis, we realize that its expression per se does not correlate with survival in our glioblastoma cohort of patients. However, it must be highlighted that our patients only received standard therapies. We can only speculate that if the PD-L1-positive patients had received an anti-PD-L1 treatment in combination with the standard of care, their survival could have improved. A clinical study testing the efficacy of anti-PD-L1 drugs for glioblastoma is warranted, and the correlation of the expression level of PD-L1 with response to such treatments should be further investigated.

## Figures and Tables

**Figure 1 biomedicines-11-00311-f001:**
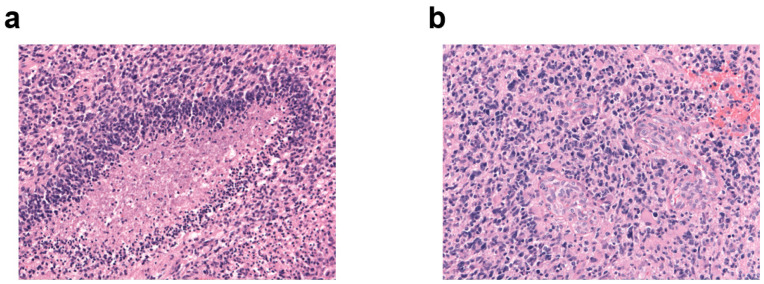
Characteristic morphology of glioblastoma. H&E immunostaining, original magnification ×20 of (**a**) palisade necrosis and (**b**) microvascular proliferation.

**Figure 2 biomedicines-11-00311-f002:**
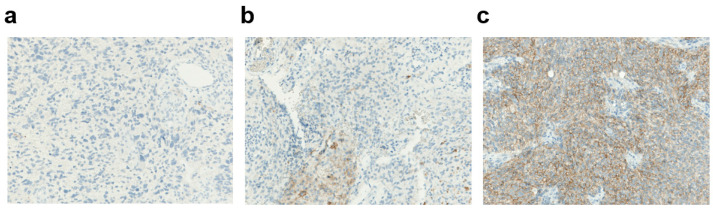
Membraneous PD-L1 expression in glioblastoma. (**a**) Absent PD-L1 expression (magnification ×40). (**b**) Intermediate PD-L1 expression (magnification ×40). (**c**) High PD-L1 expression (magnification ×40).

**Figure 3 biomedicines-11-00311-f003:**
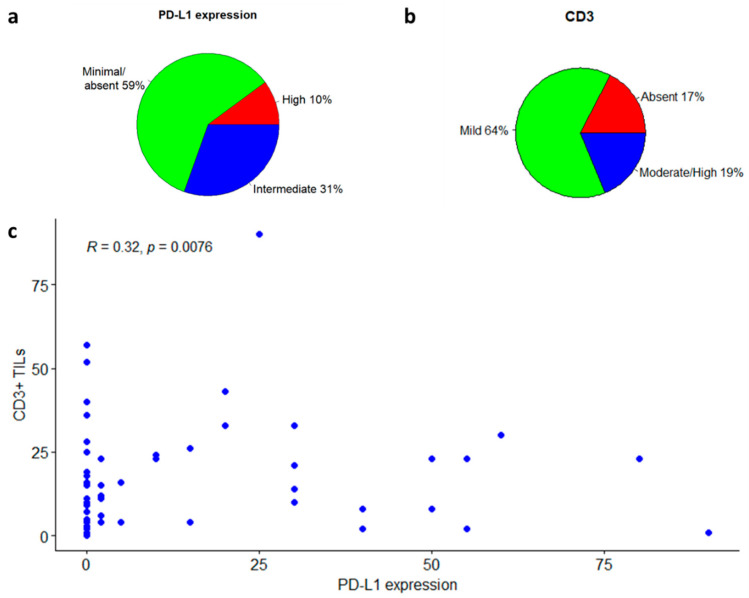
CD3 to PD-L1 correlation in the entire cohort of glioblastoma patients. (**a**) PD-L1 expression levels in glioblastoma samples. (**b**) CD3 levels in glioblastoma samples. (**c**) CD3 to PD-L1 correlation illustration.

**Figure 4 biomedicines-11-00311-f004:**
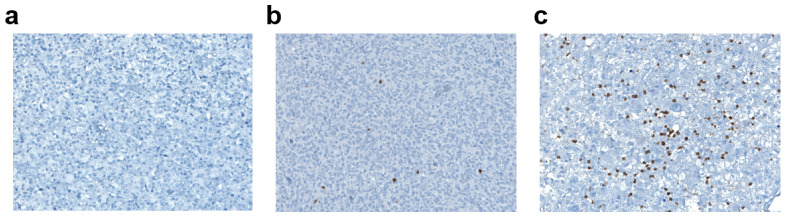
CD3+ lymphocytic infiltrate in glioblastoma. (**a**) Absent CD3+ lymphocytic infiltrate (magnification ×40). (**b**) Mild CD3+ lymphocytic infiltrate (magnification ×40). (**c**) Moderate/High CD3+ lymphocytic infiltrate (magnification ×40).

**Figure 5 biomedicines-11-00311-f005:**
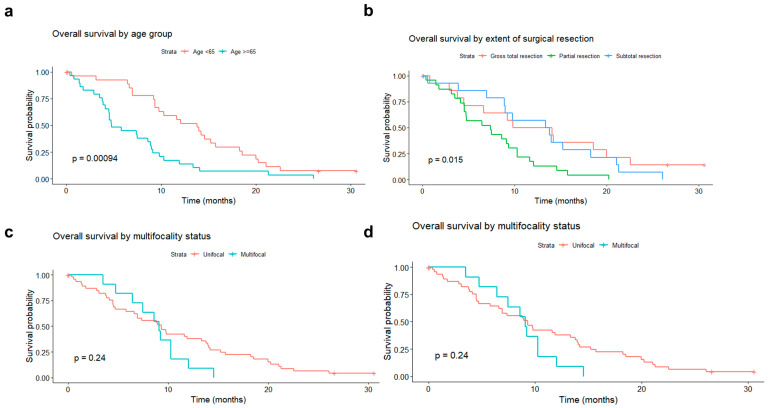
Overall survival (OS) rate of patients operated on for primary glioblastoma, according to: (**a**) age group (<65 vs. ≥65 years old); (**b**) type of surgery (gross total resection vs. subtotal resection vs. partial resection); (**c**) tumor unifocality vs. multifocality; (**d**) O6-Methylguanine-DNA methyltransferase promoter methylation status (methylated vs. unmethylated).

**Figure 6 biomedicines-11-00311-f006:**
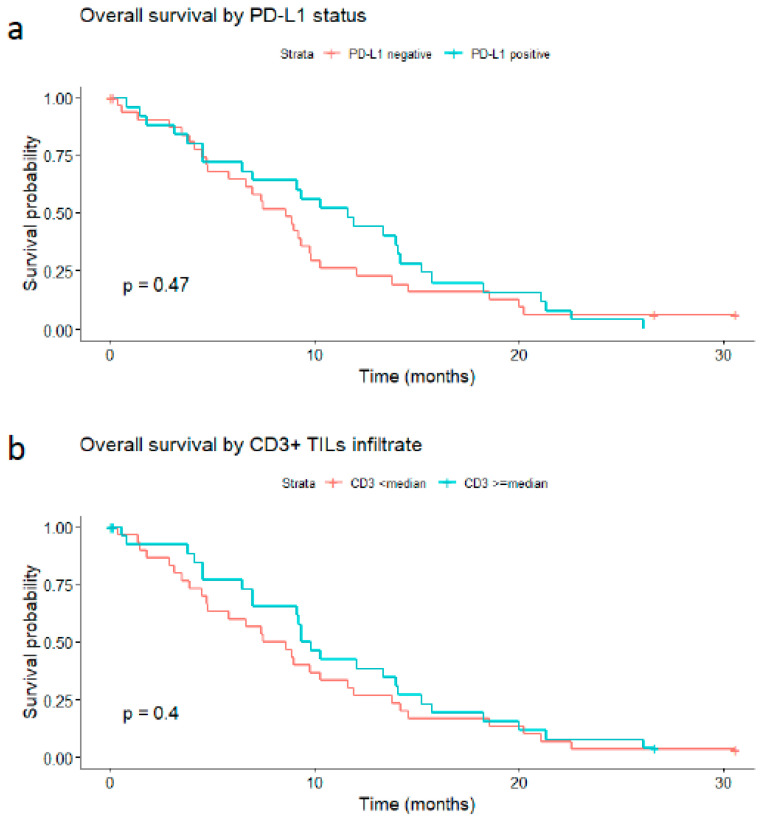
Impact of PD-L1 tumor expression and CD3-positive tumor-infiltrating lymphocytes on the overall survival rate of glioblastoma patients. (**a**) PD-L1-positive (expression in ≥1% of tumor cells) versus PD-L1-negative tumors; (**b**) high CD3 expression (≥median) versus low expression (<median).

**Figure 7 biomedicines-11-00311-f007:**
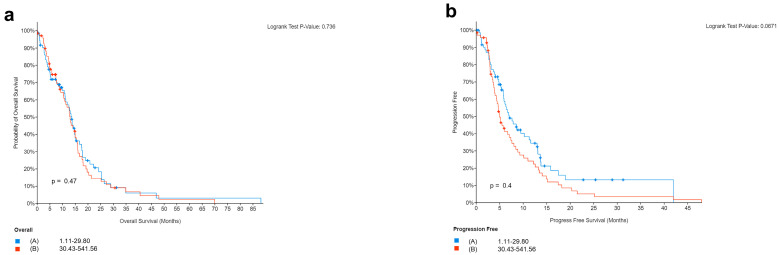
Survival in relation to PD-L1 expression in patients from the Cancer Genome Atlas (TCGA) PanCancer Atlas. Overall (**a**) and progression-free (**b**) survival in 142 patients with glioblastoma (IDH1/2 wild type), according to PD-L1 expression levels (below to above median).

**Table 1 biomedicines-11-00311-t001:** Patient and tumor characteristics.

Feature	Primary Tumors	Relapse	Overall
	N. (%) *	N. (%) *	N. (%) *
Total number	58	11	69
Age: median (range)	66 (41–81)	57 (45–78)	64 (41–81)
Sex			
Female	22 (38)	6 (55)	28 (41)
Male	36 (62)	5 (45)	41 (59)
Type of resection			
Gross total resection	15 (28)	3 (43)	18 (30)
Subtotal resection	15 (28)	1 (14)	16 (27)
Partial resection	23 (43)	3 (43)	26 (43)
N.A.	5 (-)	4 (-)	9 (-)
Alive at the end of follow up			
Yes	4 (7)	1 (9)	5 (7)
No	54 (93)	10 (91)	64 (93)
Histotype			
Glioblastoma	56 (97)	10 (91)	66 (96)
Gliosarcoma	2 (3)	1 (9)	3 (4)
Hemisphere			
Right	32 (55)	5 (46)	37 (54)
Left	26 (45)	4 (36)	30 (43)
Other **	0 (0)	2 (18)	2 (3)
Multifocal			
No	47 (81)	9 (82)	56 (81)
Yes	11 (19)	2 (18)	13 (19)
IDH1 mutated	0 (0)	0 (0)	0 (0)
MGMT promoter			
Methylated	21 (38)	7 (78)	28 (43)
Unmethylated	35 (62)	2 (22)	37 (57)
N.A.	4 (-)	2 (-)	4 (-)
Mib1: median (range)	35% (4–80%)	43% (10–60%)	35% (4–80%)
P53			
Negative	7 (12)	4 (36)	11 (16)
Intermediate	40 (69)	4 (36)	44 (64)
Positive	11 (19)	3 (27)	14 (20)
P53			
Wild type	40 (69)	4 (36)	44 (64)
Mutated	18 (31)	7 (64)	25 (36)
CD3			
Absent	10 (17)	2 (18)	12 (17)
Mild	37 (64)	7 (64)	44 (64)
Moderate/High	11 (19)	2 (18)	13 (19)
CD3+ TILs count: median (range)	8 (0–90)	11 (1–57)	9 (0–90)
PD-L1			
<1%	33 (57)	8 (73)	41 (59)
≥1%	25 (43)	3 (27)	28 (41)

N.A., not available. * Percentages are calculated within the columns of the table (Primary tumors, Relapse, Overall) and excluding the missing values. ** Other: one bilateral, one along the fourth ventricle.

**Table 2 biomedicines-11-00311-t002:** Univariate Cox models analyses of glioblastoma patients.

Variable	Hazard Ratio	95% Confidence Interval	*p*-Value *
Age	1.121	1.029–1.220	0.009
tt(Age)	0.963	0.927–0.9996	0.048
Sex			
Female	1		
Male	0.891	0.511–1.554	0.685
Type of surgery			
GTR	1		
Subtotal	1.255	0.576–2.733	0.568
Partial	2.675	1.263–5.667	0.01
Hemisphere			
Right	1		
Left	0.72	0.419–1.235	0.233
Multifocality			
No	1		
Yes	0.061	0.002–1.492	0.086
tt(Multifocality)	5.564	1.227–25.23	0.026
MGMT			
Methylated	1		
Unmethylated	0.485	0.131–1.797	0.279
tt(MGMT)	1.868	1.009–3.458	0.047
MIB1	0.996	0.976–1.016	0.674
p53			
Wild type	1		
Mutated	0.994	0.553–1.785	0.983
PD-L1			
Negative	1		
Positive	0.8197	0.478–1.407	0.471
CD3			
<median	1		
≥median	0.741	0.433–1.268	0.274

Legends: * Wald test. tt, time-dependent coefficient; GTR, gross total resection. For variables modeled with time-dependent coefficients, the ultimate coefficient can be calculated from Cox model coefficients [=log(Hazard Ratio)] as follows: fixed coefficient + tt coefficient × log (time), with time in months.

**Table 3 biomedicines-11-00311-t003:** Multivariate Cox model analyses of glioblastoma patients.

Variable	Hazard Ratio	95% Confidence Interval	*p*-Value *
Age	1.111	1.013–1.219	0.025
tt(Age)	0.97	0.931–1.010	0.143
Type of surgery			
GTR	1		
Subtotal	1.28	0.551–2.972	0.566
Partial	2.749	1.135–6.661	0.025
Multifocality			
No	1		
Yes	0.052	0.002–1.208	0.065
tt(Multifocality)	4.335	0.983–19.114	0.053
MGMT			
Methylated	1		
Unmethylated	0.652	0.158–2.695	0.555
tt(MGMT)	1.651	0.847–3.218	0.141

Legend: * Wald test. tt, time-dependent coefficient.

**Table 4 biomedicines-11-00311-t004:** Ongoing phase II and III clinical trials of ICIs.

ClinicalTrials.gov Identifier	Treatment	PHASE OF TRIAL	Primary End Point	Summary of Results
NCT02017717	Nivolumab vs. bevacizumab	III	OS	Median OS 9.5 months vs. 9.8 months
NCT02617589	Nivolumab vs. Temozolomide + radiation therapy	III	OS	Median OS 13.4months vs. 14.88 months
NCT02667587	Temozolomide + radiation therapy + nivolumab or placebo	III	PFS and OS	No survival advantage over placebo
NCT03743662	Nivolumab with radiation therapy and bevacizumab	II	OS	Ongoing study
NCT02550249	Neoadjuvant nivolumab	II	Efficacy and safety	Median OS 7.3 months
NCT04396860	Lpilimumab and nivolumab + radiation therapy	II/III	PFS and OS	Ongoing study
NCT04145115	Lpilimumab + nivolumab	II	ORR	Ongoing study
NCT02337491	Pembrolizumab with or without bevacizumab	II	MTD, DLT and PFS	Median OS 8.8 months together vs. 10.3 months for pembrolizumab alone
NCT02336165	Durvalumab monotherapy, with bevacizumab or with radiaotherapy	II	OS and PFS	Ongoing study

## Data Availability

The original contributions presented in the study are all enclosed in the article. Further inquiries can be directed to the corresponding authors.

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
