# Peer review of "Analysis of PD-L1 and CD3 Expression in Glioblastoma Patients and Correlation with Outcome: A Single Center Report"

_biomedicines, 2023, doi:10.3390/biomedicines11020311_

Round 1

Reviewer 1 Report

The manuscript entitled” Analysis of PD-L1 and CD3 Expression in Glioblastoma Patients and correlation with Outcome: a single centre report” by Navid Sobhani and collaborators is focusing on the immunological checkpoints in glioblastoma. The subject is of interest and the manuscript is nicely structured. Some concerns are listed below: 

From the Abstract section, please remove the last phrase.  

In 2.1. section, 81 FFFPE tumor tissue specimens were mentioned, yet only 72 have been actually used for the study. Please mention the exclusion criteria applied.

For the 72 FFFPE tumor tissue specimens, if possible, please attribute them to the four GBM subtypes (classical, neural, proneural, and mesenchymal). Previous studies already showed different expression of CD3 between GBM subtypes.

In 2.3.2. section, for Previously published semiquantitative evaluation, please add references.

In the 2.4. section, please mention the methods used to determine methylation status and to evaluate MIB-1 expression.

From 3.1.3. section, please remove PD-L1 was present in 28 patients (39% total, of which 29% were positive and 10% expressed it at high levels), while it was absent in 44 patients (61%). This information is already mentioned in the previous section.

In 3.2.1. section, please consider analysis the survival rate based on gender, in addition to age and tissue parameters. Also, please consider splitting the existing section in two: Correlation of age with survival and Correlation of tissue-based parameters with survival.

In 3.2.1. section, the authors discuss about p53 evaluation, but there is no mention about this marker in the Methods section. Please check.

In 3.2.2. section, it seems that some values are missing. Please check.

In 3.2.2. section, the authors mention A statistical analysis showed…. Please mention the statistical analysis or reformulate.

Figure 7 would be more valuable in the Discussion section in order to sustain the authors findings.

Please add a section with the limitations of the study. These limitations, already mentioned by the authors, are disrupting the fluency of Discussion section. Also, please avoid expressions such as It would be worth nothing in the Study limitations section.

Please check the Supplementary files. The figures S2 and S3 are missing.

Please explain the abbreviations G.B. and GTR.

Author Response

Although with controversial results, several studies have investigated the correlation between prognosis and PD-L1 and CD3 biomarkers expression. Of note, Wang et al performed a meta-analysis of 15 studies including a total of 108 patients. The outcome reported that increased PD-L1 expression can predict unfavourable OS in GBM, therefore corroborating the idea of PD-L1 as a prognostic biomarker for GBM20. Similarly, El Samman et al performed immunohistochemical analysis on 30 GBM cancer samples reporting that high PD-L1 expression was correlated with morse PFS and OS where correlation between CD3 and prognosis was statistically insignificant21. The systematic literature review and meta-analysis performed by Hao et colleagues corroborated the concept that PD-L1 is correlated with poor survival, thus exerting the role of a prognostic marker 20. Indeed, the analysis involved  9 studies for a total of 806 GBM patients. The pooled results reported a significant correlation between poor survival and PD-L1 expression.

In contrast, only a few investigations reported no significant correlation between prognosis and expression of PD-L1 and CD3 markers. Among these, Berghoff et al assessed CD3 and PD-L1 expression in 135 GBM samples using an immunohistochemistry assay. Although PD-L1 and CD3 markers were significantly detectable in the vast majority of GBM, these were not correlated with prognosis22. Ultimately, we reported the study by Dede et colleagues where 90 GBM cancer samples were immunohistochemically investigated. A slighlty positive correlation was found between CD3 expression and better prognosis whereas no significant correlation was found between PD-L1 expression and worse prognosis23.

Reviewer 2 Report

The design and result of the study do not support the objective of this paper.  The author claimed that CD3 and PDL1 could serve as a biomarker for selecting patients that will respond to immunotherapy.  But this paper did not include the study of GBM patients and their treatment course and correlate with the levels of CD3 and PDL1.  The author found that the majority of GBM patients expressed a high co-expression of both CD3 and PDL1 can be used as a biomarker for immunotherapy treatment, and correlate these biomarkers with their survival rate.  They then conclude that the survival of GBM patients is not associated with the expression of either CD3 and PDL1.   But there is no information about the immunotherapy treatment that these GBM patients received to correlate with their survival rate and the effectiveness of the treatment and the levels of different biomarkers studied.  The author could revise this paper to focus on the study of the tumor immune microenvironment rather than claiming the potential of CD3 and PDL1 or other biomarkers as predictive biomarkers in responding to immunotherapy.  More comments the attached.

Author Response

Reviewer 2:

The design and result of the study do not support the objective of this paper.  The author claimed that CD3 and PDL1 could serve as a biomarker for selecting patients that will respond to immunotherapy.  But this paper did not include the study of GBM patients and their treatment course and correlate with the levels of CD3 and PDL1.  The author found that the majority of GBM patients expressed a high co-expression of both CD3 and PDL1 can be used as a biomarker for immunotherapy treatment, and correlate these biomarkers with their survival rate.  They then conclude that the survival of GBM patients is not associated with the expression of either CD3 and PDL1.   But there is no information about the immunotherapy treatment that these GBM patients received to correlate with their survival rate and the effectiveness of the treatment and the levels of different biomarkers studied.  The author could revise this paper to focus on the study of the tumor immune microenvironment rather than claiming the potential of CD3 and PDL1 or other biomarkers as predictive biomarkers in responding to immunotherapy.  More comments the attached:

We have fixed the paper. We do not say that CD3 and PD-L1 could be biomarkers for immunotherapy. We concluded now that in glioblastoma PD-L1 levels do not correlate with survival. Also, that CD3 level do not correlate with survival.

This paper investigated the level of co-expression of CD 3 and PD-L1 expression as a biomarker for GBM patients using immunostaining of the tumor tissues from patients whom were newly diagnosed with GBM. The co-expression of a high level of CD3 and PDL1 was observed in the majority of GBM tumor tissues. However, this paper fails to conclusively correlate/associated a high level of CD3 and PDL1 to a high response of GBM patient to immunotherapy/immune checkpoint inhibitor treatment. There is no information about the treatment that these GBM patients received to correlate to their OS rate.

We have fixed it and clearly explained that the expression levels of PD-L1 and CD3 do not correlate with survival of patients treated with standard therapy. Our was not a study or clinical trial relating to immunotherapies in GBM.

Comments

Lots of typo and unnecessary capitalizing of non-special terms throughout this paper. Please send the paper for an English editing service prior to submitting for publication.

Alberto D’Angelo, who has graduated from UK, has proofread the paper and fixed the typos.

There are a few sentences that is long, windy and difficult to understand. “However, Central Nervous System (CNS) has recently lost its connotation of being an immunologically privileged site thanks to the discovery of resident macrophages, lymphatic vessels alongside the dural sinuses and active interaction between CNS and peripheral immune cells.”

We have fixed it.

Reference citation formatting is also not correctly done and not consistent throughout the manuscript.

We have fixed it.

Introduction of GBM biomarkers needs to be further expanded. The study of GBM and its TME and immunotherapy for GBM has been widely investigated. The author should comment if CD 3 and PDL1 are biomarkers that have been identified by other investigator. The author should please elaborate on the novelty of this study. How is this study different compared to the published paper.

We mentioned the literature in the introduction:

A high PDL1 expression is often associated with an immunosuppressive TME, which is being acknowledge by the author as well. Authors should also include in the introduction and discussion section on the association of PDL1 expression to the pathological response of GBM patients to immunotherapy. This paper claims the importance seeking for GBM biomarkers to improve the immunotherapy treatment to the majority of the GBM patients. However, the result performed in the paper does not support the author hypothesis.

We have fixed it. We have mentioned that PD-L1 expression could be an indicator of immunotherapy response, however, since there is not a correlation with survival, the potential of a benefit from immunotherapy should be wisely measured.

Can author provide more details how the result of this studies (high expression of both high CD3 and PDL1) be applied in clinical application? How does a physician obtain the levels of CD3 and PD-L1 in GBM patient prior to receiving immunotherapy.

We have fixed it. Because our re-analysis showed that there is not such a correlation, we decided to remove this statement. The statistical analyses are more stringent now. We preferred to publish negative results.

CD3 is a protein that activate CD4 and CD8 T-cells. Thus, a high level of CD3 is associated with active T- cells. The author should test the level of immmunoactivator cytokines ie.: TNF-γ in order to confirm the cytotoxicity activity of T-cells.

As just mentioned, we did not see any correlation between immune markers and survival. Therefore, there is no need to confirm the cytotoxicity activity of T cells.

Round 2

Reviewer 1 Report

Minor concerns:

Please check the references in the Introduction Section. Some of them (e.g. 1, 6, 8) appear to be in a different format.

Please check Section 2.4. MGMT and MIB-1. Please adjust the title of the section or move the information regarding p53 elsewhere.

Please check Section 2.5. p53 and remove it seems that some values are missing.

Please check Section 3.1.1. Patients. Considering that in section 2. Patients and Materials is directly stated that 69 patients were involved in this study, please remove Among 81 patients operated between 2018 and 2020, a review of histological specimens confirmed a diagnosis of glioblastoma according to the 2021 revision of the World Health Organization Classification of Central Nervous System Tumors (IDH-wildtype, grade 4) in 69 cases. Three cases were excluded because they had IDH1 mutation and additionally 9 cases were excluded because the tissue samples were assessed as quantitatively or qualitatively insufficient for analysis.

Considering that the authors moved Figure 7 in the Discussion Section, the text concerning that figure should be also moved in the Discussion Section.

Please add a Conclusion Section to include the conclusion of the study that is already mention as the final paragraph of Discussion Section.

Author Response

Revewer 1

Please check the references in the Introduction Section. Some of them (e.g. 1, 6, 8) appear to be in a different format.

fixed

Please check Section 2.4. MGMT and MIB-1. Please adjust the title of the section or move the information regarding p53 elsewhere.

Fixed.

Please check Section 2.5. p53 and remove it seems that some values are missing.

Fixed.

Please check Section 3.1.1. Patients. Considering that in section 2. Patients and Materials is directly stated that 69 patients were involved in this study, please remove Among 81 patients operated between 2018 and 2020, a review of histological specimens confirmed a diagnosis of glioblastoma according to the 2021 revision of the World Health Organization Classification of Central Nervous System Tumors (IDH-wildtype, grade 4) in 69 cases. Three cases were excluded because they had IDH1 mutation and additionally 9 cases were excluded because the tissue samples were assessed as quantitatively or qualitatively insufficient for analysis.

Ok.

Considering that the authors moved Figure 7 in the Discussion Section, the text concerning that figure should be also moved in the Discussion Section.

Moved.

Please add a Conclusion Section to include the conclusion of the study that is already mention as the final paragraph of Discussion Section.

Added.

Reviewer 2 Report

Central Nervous System, Tumor-Infiltrating Lympocyte should not be capitalized.  These are not special names

The format of citation (in text) is inconsistent.  The first page in-text citation is different than the rest of the manuscript. 

End of pg 2 quotes "the co-expression of either CD3 or PD-L1 correlated with improved overall survial in the GBM".  This again is contradictory with the results of the study. 

Section 3.2.2 ....it was .(??) .months (??)

Discussion section...typo "habe"

Discussion pg 11 PD1-PDL1 "actove" axis?

Discussion pg 10 last sentence ....showed statistical significance of (what)

Discussion pg 11. Other works suggest that PDL1 is not a predictive biomarker.  And the authors also conclude similar results that neither PDL1 and CD3 is not a predictive biomarkers of survival.  But the last sentence of the paragraph ends with "PD-L1 biomarker is a useful biomarker for predicting patients would best respond to anti-PDL1 or antiPD1 therapies"     Perhaps the author can elaborate on the differences in patients' "best response" to ICB and their OS.  It is not clear to me what is the distinction between treatment response to survival.  Patients with better treatment response should results in higher OS rate.  

Author Response

Reviewer 2

Central Nervous System, Tumor-Infiltrating Lympocyte should not be capitalized.  These are not special names

Fixed.

The format of citation (in text) is inconsistent.  The first page in-text citation is different than the rest of the manuscript. 

Fixed.

End of pg 2 quotes "the co-expression of either CD3 or PD-L1 correlated with improved overall survial in the GBM".  This again is contradictory with the results of the study. 

We have removed it.

Section 3.2.2 ....it was .(??) .months (??)

We have fixed in this new version. “In the PD-L1 negative group (33 patients), the median OS was 8.6 months. In the positive one (25 patients), it was 11.6 months.” There was a difference, but it was not statistically significant.

Discussion section...typo "habe"

We did not find that typo in the new version that we have fixed. It was a German keyboard of one colleagues probably.

Discussion pg 11 PD1-PDL1 "actove" axis?

Fixed.

Discussion pg 10 last sentence ....showed statistical significance of (what)

We have changed that last conclusive sentence in the new version of the paper we have previously sent. Also we made it now into a conclusions. We have fixed also a previous similar statement.

Discussion pg 11. Other works suggest that PDL1 is not a predictive biomarker.  And the authors also conclude similar results that neither PDL1 and CD3 is not a predictive biomarkers of survival.  But the last sentence of the paragraph ends with "PD-L1 biomarker is a useful biomarker for predicting patients would best respond to anti-PDL1 or antiPD1 therapies"     Perhaps the author can elaborate on the differences in patients' "best response" to ICB and their OS.  It is not clear to me what is the distinction between treatment response to survival.  Patients with better treatment response should results in higher OS rate. 

We have added a statement in the conclusions explaining how this could have happened and how using an immunotherapy next time could improve the survival of the PD-L1 positive population of patients too. “Moreover, since PD-L1 has been used in solid tumors as a valid biomarker for potential response to immune checkpoint blockade against the PD-1/PD-L1 axis, we realize that its expression per-se does not correlate with survival in our glioblastoma cohort of patients. However, it must be highlighted that our patients only received standard therapies. We can only speculate that if the PD-L1 positive patients would have received an anti-PD-L1 therapy in combination with the standard of care, their survival could have improved significantly.”

Round 3

Reviewer 2 Report

There are still a few more typos in the revised version.  Suggest the author to fix it.  Other than that, the author has addressed all of my concerns regarding this paper.  

Author Response

Dear Reviewer,

Thank you. We have corrected the minor typos and checked it with Grammarly. We will continue to do so during the proofs.

Regards,

Navid Sobhani (corresponding author)